PluDG: enhancing task-oriented dialogue system with knowledge graph plug-in module

Dong Xuelian
Chen Jiale jalorc@163.com
School of Computer Science, University of South China , Hunan , China
Nguyen Binh
Electronic publication date: 2023 Nov 24
Publication date: 2023
Volume: 9
Electronic Location ID: e1707
Received 2023 Sep 8; Accepted 2023 Oct 27
Copyright: ©2023 Dong and Chen
Copyright year: 2023
Copyright holder: Dong and Chen
License: This is an open access article distributed under the terms of the Creative Commons Attribution License, which permits unrestricted use, distribution, reproduction and adaptation in any medium and for any purpose provided that it is properly attributed. For attribution, the original author(s), title, publication source (PeerJ Computer Science) and either DOI or URL of the article must be cited.
License URL: https://creativecommons.org/licenses/by/4.0/

Keywords: Artificial intelligence, Natural language processing, Data science, Graph neural networks, Dialogue systems

Funding: The authors received no funding for this work.

==============================
Task-oriented dialogue systems continue to face significant challenges as they require not only an understanding of dialogue history but also domain-specific knowledge. However, knowledge is often dynamic, making it difficult to effectively integrate into the learning process. Existing large language model approaches primarily treat knowledge bases as textual resources, neglecting to capture the underlying relationships between facts within the knowledge base. To address this limitation, we propose a novel dialogue system called PluDG. We regard the knowledge as a knowledge graph and propose a knowledge extraction plug-in, Kg-Plug, to capture the features of the graph and generate prompt entities to assist the system’s dialogue generation. Besides, we propose Unified Memory Integration, a module that enhances the comprehension of the sentence’s internal structure and optimizes the knowledge base’s encoding location. We conduct experiments on three public datasets and compare PluDG with several state-of-the-art dialogue models. The experimental results indicate that PluDG achieves significant improvements in both accuracy and diversity, outperforming the current state-of-the-art dialogue system models and achieving state-of-the-art performance.

Introduction

Building task-oriented dialogue systems has become a prevalent research subject in both academic and business settings. The commonly used method to create a dialogue system is developing an end-to-end system, which increases efficiency by generating responses directly from a knowledge base and dialogue history (Lu et al., 2023a; Lu et al., 2023b; Liu et al., 2023b). Figure 1 depicts the whole data needed by the task-oriented dialogue system.

Figure 1 Illustration of a task-oriented dialogue system about navigation.

To make full use of the external knowledge base information, Madotto, Wu & Fung (2018) proposed Mem2Seq. The model enhances the MemNN framework (Sukhbaatar et al., 2015) using a sequence generation framework and incorporates a global multi-hop attention mechanism to replicate words directly from the dialogue history or knowledge base. In addition, some researchers propose that entities’ relationships in an external knowledge base should be considered rather than treated as isolated triples. Banerjee & Khapra (2019) achieved state-of-the-art results in goal-directed dialog systems using GCN (Kipf & Welling, 2016) to combine structural information with encoded sequences and developed contextual graphs for constructing hybrid dialogues in different languages. Later, Zhao et al. (2023) proposed a multi task learning method based on graph attention networks for modeling a multi-domain task-oriented dialogue system.

On the other hand, researchers have also utilized large language models (LLMs) in task-oriented dialogue systems by treating the response as the natural language generation task. One such system is UBAR (Yang, Li & Quan, 2021), a modularly designed task-based dialogue system based on GPT-2 that facilitates module replacement and functional extensions for different domains and scenarios. Rony, Usbeck & Lehmann (2022) proposed DialoKG, a model that incorporates knowledge into the GPT-2 architecture. To achieve this, the model leverages the structural information of the knowledge base by treating each entity as a sequence and calculating its weight for the dialogue history with the help of RoBERTa (Liu et al., 2019). Nevertheless, LLMs may face challenges in capturing these structured relations when processing knowledge bases to treat entities as sequences since the information contained in knowledge bases is usually structured, consisting of entities and their relations (Shen et al., 2021; Liu et al., 2023a).

To address this limitation, this article presents a novel method called PluDG (PLUgins-Assisted Dialogue Generation). Specifically, we designed a plug-and-play module called Kg-Plug, which treats knowledge as a knowledge graph. Kg-Plug utilizes LR-GCN modules to leverage low-dimensional decomposition for feature extraction. Furthermore, it employs the attention mechanism to align with the dialogue history to get the prompt entities, which are inferred from the dialogue history and knowledge base and are related to the user’s true intent. Subsequently, prompt entities are generated and provided to the decoder for dialogue generation. Additionally, we employ a GPT-2-based decoder for generating responses. We enhance it by incorporating an entity memory ensemble embedding, which utilizes special tokens and embeddings to improve GPT-2′s ability to produce contextually appropriate results.

Our article outlines several major contributions:

• We proposed PluDG, a task-oriented dialogue system that integrates a plug-and-play Kg-Plug component into a GPT-2-based decoder. PluDG learns intrinsic graph structure information from the knowledge base and gets entity hints to pass to the decoder for better response generation.

• We proposed a novel embedding technique for GPT-2, named Unified Memory Integration (UMI), which utilizes multi-layered and position embeddings that are aware of the structure of the dialogue history, knowledge base, and prompt entities.

• Experiment results on three benchmark datasets show the superior performance of PluDG compared to other state-of-the-art models. Our model outperforms existing approaches based on metrics, particularly in complex knowledge-base information datasets.

Related Works

A task-oriented dialog system has been employed with an end-to-end approach. Originally, researchers considered the KB and dialogue history as sequences. Lately, many researchers have emphasized the importance of preserving the connection between entities in the KB to achieve improved bot responses. The most recent studies have applied pre-trained language models to enhance dialog systems.

RNN-based dialogue systems. Wen et al. (2016) proposed a web-based task-oriented dialogue system capable of directly learning parameters from raw data. Later, Wu, Socher & Xiong (2019) proposed GLMP that integrates the external knowledge base. The external knowledge utilized an end-to-end memory network (MN), storing word-level information about the knowledge base and conversation history. Regrettably, prior studies have failed to acknowledge the plentiful structural information present in knowledge bases, specifically the graph structural information formed by entity-entity relationships.

Knowledge graph-augmented dialogue systems. Graph neural networks are also used by some researchers to encode knowledge-base entities. He et al. (2020) developed Fg2Seq, which can integrate the latent semantics of conversation history, improving the description of entities and enabling better inference of knowledge related to conversation history. Wu, Harris & Zhao (2022) employed a GMN to comprehend the intrinsic patterns in the dialogue history and their connection with the KB. Although this method treats the KB as a graph, their decoders are still based on RNN, which does not provide a superior understanding of contextual information compared to the GPT.

Pretrain-language-model-based dialogue systems. Madotto et al. (2020) employed a strategy called knowledge embedding to embed knowledge bases directly into model parameters. This approach does not require dialogue state tracking or template responses as inputs and can dynamically update its knowledge base through fine-tuning. Recently, Huang, Quan & Wang (2022) proposed a task-oriented dialog model that employs an Auto-regressive Entity Generation technique, which consists of three major components: a GPT-2 that generates replies, an entity generator that identifies entities in the responses, and a final stage that embeds the entities to generate the ultimate dialog response. It is an end-to-end task-oriented dialogue model that combines natural language processing and generation methods.

In contrast to previous studies, our work introduces PluDG, a novel task-oriented dialogue system. PluDG incorporates Kg-Plug, a plug-and-play component, to extract features from the knowledge base and align them with the dialogue context before passing prompt entities to the decoder. Additionally, to enhance the decoder’s comprehension of the underlying semantic information, we employ the UMI module to provide the structure of the knowledge base and dialogue history.

Method

Prior to presenting the complete method, we provide a description of the problem.

For the given dialogue history, we regard the utterance of the user as U and the system’s response as S. For given turn i, dialogue consists of Ti, which is made up of Ui and Si: Ti = (Ui, Si). If we assume that there have been K turns in the dialogue history, then the entire history can be defined as T = [T1, T2, T3, …, TK].

Regarding the knowledge base, we utilize the triple format G = (e, r, o) to represent various entities and their relationships. Note that, e refers to the entities, r represents the relationships, and o represents objects. For instance, in the case of the ith potential triple Gi, Gi = (jrestaurant, place, north).

Suppose there are n entities for a given turn i, then we use Ki to denote the given knowledge base construct by the format above mentioned Ki = (G1, G2, …, Gn).

The probability distribution of responses generated by the language model in the ith turn is formally defined as follows: (1) PSi|T1:i−1,Ui,Ki= ∏j=1NPsj|s1:j−1,T1:i−1,Ui,Ki,

where Si = [s1, …, sn] represents the response generated from the ith round of the system, and N is the maximum number of words in the response Si. The 1:j − 1 represents elements 1 to j − 1.

Overview

To address the problem that LLMs may face challenges in capturing these structured relations when processing knowledge bases, treat entities as sequences. We propose a model called PluDG. This model is composed of three components: the Kg-Plug and the Decoder. More details are shown in Fig. 2.

Figure 2 Overview of the architecture of the PluDG model.

Kg-Plug module

The Kg-Plug module is designed as a plug-and-play component, as illustrated in Fig. 3. It treats the provided knowledge as a graph and employs LR-GCN for feature extraction. Subsequently, it infers the most probable entity hints based on the dialogue history information. Finally, the prompt entities are passed to the decoder.

Figure 3 The schematic diagram of the Kg-Plug module includes the utterance encoder, context knowledge encoder, and entity reasoner components.

Utterance encoder

Assuming that there are K turns in the dialogue history, the history contains 2K − 1 utterances, where each utterance includes Li words. The words in the ith utterance are represented by word wil, where L ∈ [1, Li]. First, a Bi-GRU, which includes both a forward unit and a backward unit, is used to obtain the hidden representation of the sentences: (2) Hi=hi1,hi2,hi3,…,hil=BiGRUEmbwil,

where Emb(wil) represents the embedding state of the word wil.

Next, a self-attention unit is utilized to capture the contextual information of each token in order to obtain a comprehensible semantic representation of the utterance, as shown below: (3) μil= tanhWwhil+bw,

(4) αil=expμiluw∑l expμiluw,

(5) vi= ∑lαilhil,

where Ww, bw, uw are trainable parameters of the model.

Lastly, a GRU is utilized to encode the utterance vector vi: (6) Hic=GRUvi,i∈1,2K−1.

Context knowledge encoder

The Context Knowledge Encoder is employed to extract hidden information from both the dialogue history and knowledge base.

Context-KB Alignment. Following Chen et al. (2017), the Context-KB Alignment module aims to capture the alignment representation of each entity in the knowledge base through the incorporation of dialogue history. To achieve this goal, an attention mechanism is employed to align the dialogue history embedding with the knowledge base entity embedding, allowing for the creation of a coherent representation of the graph. Specifically, the module concatenates each word wil with the entity representation e, applies a tanh activation, and derives attention scores through a Softmax operation. These scores are then multiplied with the corresponding words and summed to generate an aligned representation of the entity’s conversation history: (7) cil= tanhWeEmbe;Embwil+be,

(8) αil=expcilue∑l expcilue,

(9) falignie= ∑lαilEmbwil,

where We, be, and ue are trainable weight parameters, and [; ] denotes the concatenation.

Next, the jth entity embedding Emb(ej) is concatenated with its correspondingly aligned embedding faligniej. In this way, we obtain a sequence of history-alignment entity input representations. Then, the sequence is passed to the GRU unit to obtain a more robust history-alignment entity representations. Formally, for each entity ej, the representation fij is obtained as follow: (10) fij=GRUEmbej;faligniej.

Knowledge Graph Encoder. In this section, we introduce a GCN (Kipf & Welling, 2016) to extract the intrinsic features of the knowledge graph. However, inspired by Hu et al. (2021), we leveraged the low-rank decomposition into the weights of GCN and named this new module LR-GCN. For given weights W0 ∈ ℜx×y, we use Wo + ΔW = Wo + BA to replace the update, where B ∈ ℜx×y, A ∈ ℜy×z, and y <  < min(x, z).

In this section, we represent each entity as a node, where N represents the set of nodes. The relationships between entities are denoted as edges, and R represents the set of edges. Following the Context-KB Alignment operation, each entity in the dialog history has 2K − 1 representations, which correspond to the 2K − 1 utterances spoken. To capture the features from each node and its neighborhoods, we employ the GCN in the Graph operation: (11) gij=σ∑r∈R ∑v∈Nir1|Nir|Wrfiv+W0fij.

In Eq. (11), Nir denotes the set of neighborhood-indices of entity i under relation r, r ∈ R; Wr and Wo are trainable parameters. An activation function σ() is adopted in this research, and ReLU is the specific function utilized.

Finally, an appropriate pooling method is used to fuse the data in gij and fij to obtain a question and text representation matrix Gf: (12) ϑij= tanhWgfij;gij+bg,

(13) αil=expϑijug∑l expϑijug,

(14) gif= ∑lαilfij;gij,

where Wg, bg, and ug are trainable weight parameters, [; ] denotes the concatenation, and Gf=g1f,…,g2K−1f.

Entity reasoner

The entity reasoner is an important component of the Kg-Plug. In this component, we concatenate the Utterance Encoder’s output and Context Knowledge Encoder’s output as q0r, formally: (15) q0r=Hc;G,

then use to two-hop attention to get the final entity probability.

Two-Hop update. In the reason stage, followed by the MemNN, we design a two-hop update mechanism to get the precise entity. For the sake of clarity in our description, we denote the number of hops as X, where X = 2. For the given hidden state, q0r, we use learnable attention to search for deeper information. In each hop have the following: (16) qi+1r= tanhWqqir,i∈0,X.

In the final hop, we use the Softmax function to get the final entity probability pent: (17) pent=SoftmaxGTWeqXr.

Decoder

In this study, the decoder is based on the GPT-2 model and is responsible for generating the final response.

Unified Memory Integration. As shown in Fig. 4, to incorporate entity structural information from the Knowledge base and prompt entities from Kg-Plug into the GPT-2, we use various embedding techniques, including entity embedding, type embedding, as well as the traditional word token and positional embedding. These techniques enable the decoder to extract the knowledge graph structure, which is linearized into a sequence as input, with special tokens ([NAME] and [ADDR] etc.) to separate the subject, relation, and object of an entity. The entity embedding layers capture entity-level separate information about the word token, and the type embedding distinguishes the relevant tokens. Furthermore, we incorporate speaker information into the dialogue history. To differentiate between the system’s response and the user’s utterance, we employ the [SYS] token for system responses and the [USR] token for user utterances. Additionally, we use [Query] to indicate the user’s current utterance for clear separation.

Figure 4 Illustration of Unified Memory Integration.

For generating responses, the GPT-2 decoder relies heavily on the input sequence, and the sequence of tokens plays a crucial role in determining the output. We position the prompt entities after the history, as shown in Fig. 4, in order to enhance the generation process. By doing so, we hope the decoder can draw upon a more precise context, which improves its ability to understand user queries and generate appropriate responses.

Calculate the modeling response word’s probability pfinal by using the embedding token as follows: (18) h0t=ex′Wv+Wp,

(19) hlt=TransformerBlockhl−1t,

(20) pfinal=SoftmaxhltWv,

where ex′ presents x′ in one-hot representation, Wv presents the word vector mask, Wp is the position mask, and l ∈ L presents the Transformer layers.

Experiments

Datasets

We evaluate our model on three publicly available benchmark datasets: CamRest (Wen et al., 2016), In-Car Assistant (Eric & Manning, 2017), MultiWOZ 2.1 (Budzianowski et al., 2018). Details of each dataset are provided below:

• CamRest. The dataset comprises dialogs in the restaurant reservation domain, consisting of 676 multi-turn dialogs with an average of five turns per dialog. Additionally, each dialog has an average of 22.5 KB of triples. To conduct our experiments, following Rony, Usbeck & Lehmann (2022), we partitioned the dataset into training, validation, and test sets, with 406, 135, and 135 dialogs, respectively.

• In-Car Assistant. The dataset contains 3,031 multi-turn dialogs across three distinct domains: weather, navigation, and schedule. On average, each dialog comprises 2.6 turns, but the knowledge base (KB) information in every dialog has an average of 62.3 triples. Following Rony, Usbeck & Lehmann (2022), we partitioned the In-Car Assistant dataset into training, validation, and test sets, consisting of 2,425, 302, and 304 dialogs, respectively, for use in our experiments.

• Multi-WOZ 2.1. The dataset comprises three distinct domains: attractions, hotels, and restaurants. Each dialog in the dataset has an average of 5.6 turns and 54.4 KB of triples. To process the data, we followed the method used by Rony, Usbeck & Lehmann (2022) and divided the dataset into training, validation, and test sets, each containing 1,839, 117, and 141 dialogs, respectively.

Baselines

For our PluDG model, we employ some of the recently proposed state-of-the-art models, including GLMP (Wu, Socher & Xiong, 2019), DF-Net (Qin et al., 2020), Fg2Seq (He et al., 2020), GPT-2+KE (Madotto et al., 2020), CDNet (Raghu et al., 2021), GraphMemDialog (GMD) (Wu, Harris & Zhao, 2022), and DialoKG (Rony, Usbeck & Lehmann, 2022). All comparison models were evaluated in the same experimental environment.

Metrics

We utilize two popular evaluation metrics in dialogue studies to evaluate our model: BLEU (Papineni et al., 2001) and Entity F1. To ensure a fair comparison with previous work, we adopted these widely used metrics in the community.

• BLEU. The Bilingual Evaluation Understudy (BLEU) metric measures the n-gram overlap between generated responses and gold standard responses.

• Entity F1. We use Entity F1 to assess the system’s ability to produce relevant entities that can accomplish specific tasks by retrieving accurate entities from the provided knowledge base. To compute the Entity F1 score, we micro-average the precision and recall over knowledge base entities of the generated responses.

Model training

The cross-entropy is utilized to direct the model-training process. Specifically, the negative log likelihood is calculated between the predicted and actual distributions of the training data: (21) LD=−∑j|D|∑in logpsij|s1:ij,T,K,

where D is the dialogue dataset consisting of D1, D2, …, Di.|D| is the number of the dialogue datasets. Let sij be the response generated by the model at Dj, corresponding to the words output by the ith time step of the model. Here, n represents the maximum response length, while dialogue history T and knowledge base K are given by Dj.

Training settings

We employed the PyTorch framework to implement our model, which was trained on an NVIDIA GeForce GTX 3070 with 8 GB of GPU memory. Our experiments entailed setting the Kg-Plug’s embedding dimensions and hidden units to 128, while the batch size was set to 8. Additionally, we set the number of hops for the Entity Reasoner at 2.

For the decoder, we used the normal pretrained GPT-2 with 137M parameters. The model underwent end-to-end training utilizing the AdamW23 optimizer, with the learning rate was set to 6.25e−5 and the decay was set to 1e−8. For all the datasets, the dropout ratio was set at 0.2. More hyper-parameters used to train PluDG are listed in Table 1.

Table 1 Training parameters.

Parameter	Kg-Plug	GPT-2+UMI+Kg-Plug	
Batch size	8	2	
Learning rate	1e−4	6.25e−5	
Epoch	20	10	
Dropout	0.2	0.2	
Embedding size	128	768	
Max gradient norm	1	1	

Evaluation results

Table 2 illustrates the superior performance of our model compared to baseline models on three datasets, as demonstrated by both BLEU (Papineni et al., 2001) and Entity-F1 metrics. Additionally, we present the architectures the models utilized. Our experimental results indicate that PluDG achieves a BLEU score of 23.0 and an F1 score of 76.9 on the CamRest dataset, along with a significantly improved BLEU score of 21.6 and 69.5 Entity-F1 score on the In-Car Assistant dataset, showcasing its capability to generate more fluent responses. Moreover, we achieve a higher Entity-F1 score of 42.4 on WOZ2.1, despite obtaining a BLEU score of 9.2. Notably, PluDG outperforms the previously similarly structured DialoKG (Rony, Usbeck & Lehmann, 2022) model in different domains, highlighting the effectiveness of Kg-Plug in constructing knowledge graph extraction features and providing effective prompt entities. Additionally, UMI modules effectively leverage deep semantic information, further contributing to the model’s response efficacy. The same trend of improvement is observed in three datasets, indicating the generalization ability of our model.

Table 2 Comparison of generation results on three datasets.

Model	Structure	CamRest	In-Car Assistant	MultiWOZ 2.1	
		BLEU	Entity F1	BLEU	Entity F1	BLEU	Entity F1	
GLMP	RNN	–	–	8.5	58.4	–	–	
DFNet	RNN	–	–	9.00	62.7	3.4	34.8	
FG2Seq	RNN+GCN	13.2	62.2	10.4	62.0	–	–	
GPT-2+KE	GPT-2+KE	17.8	54.0	16.8	58.6	12.7	35.6	
CDNet	DNN	19.1	63.1	16.0	57.4	10.5	30.6	
GMD*	GMN+GAT	22.3	64.4	18.8	64.5	14.9	40.2	
DialoKG	GPT-2+RoBerta	22.5	75.4	18.4	64.9	7.4	39.1	
PluDG (Ours)	GPT-2+Plug-in	23.0	76.9	21.6	69.5	9.2	42.4	

Ablation study

To assess the necessity of each component in PluDG, we conducted an ablation study by removing the Kg-Plug and Unified Memory Integration (UMI) modules and analyzing their impact on the performance of the framework. As shown in Table 3, our results indicate that these two modules are essential for achieving high performance in task-oriented dialogue generation tasks.

Table 3 Results of the ablation study.

Model	CamRest	In-Car assistant	MultiWOZ 2.1	
	BLEU	Entity F1	BLEU	Entity F1	BLEU	Entity F1	
PluDG	23.0	76.9	21.6	69.5	9.2	42.4	
w/o Kg-Plug	22.4	73.2	20.3	64.7	8.8	40.5	
w/o UMI	22.8	75.2	6.3	72.7	8.2	41.9	
w/o Both	21.7	74.6	18.2	64.8	7.4	39.0	

After removing Kg-Plug, a component added as a plug-in to the model, we observed a significant drop in performance for various evaluation indicators, particularly Entity F1 of CamRest and In-Car Assistant, both decreasing by more than three points. We speculate that the Prompt entities provided to the GPT-2 decoder play a vital role in generating responses. Conversely, removing the UMI module leads to a performance drop across all the three datasets. Although the BLEU index of the In-Car dataset experienced the most significant drop, exceeding 15 points, the Entity F1 indicator increased. Thus, we conjecture that the GPT-2 model heavily relies on input sequences for response generation, and labeled information significantly impacts the output. By incorporating more semantic information, the GPT-2 model obtains a more accurate and comprehensive context, leading to more relevant responses. Additionally, when all the extra modules were removed, we observed a drop in all indicators, performing even worse than the previous baseline model. In conclusion, our ablation study emphasizes the critical importance of Kg-Plug and UMI in PluDG, as they are essential for achieving state-of-the-art performance in task-oriented dialogue tasks.

Significance test

To rigorously assess the significance of the performance improvement in our proposed method, we conducted an evaluation using the t-test method. We compared PluDG with the best model. The comparison is divided into BLEU significance, Entity F1 significance, and the significance comparison of both. The results, presented in Table 4, demonstrate that our PluDG exhibits significantly improved performance metrics compared to the best baselines, with all p-values below the 0.05 significance level.

Table 4 Result of the significance test.

Metrics	t-statistics	p-value	
BLEU	3.7871	0.0193	
Entity F1	5.8948	0.0042	
Both	3.5881	0.0049	

Comparison with other GNN models

Our proposed approach, PluDG, exhibits significant improvements over existing baselines. We hypothesize that this improvement can be traced to the Kg-Plug for the powerful graph feature extraction. To test our hypothesis, we compared four different GNNs, including GIN24, GSE25, and GAT26, all of which were modified to directly replace our original LR-GCN modules, ensuring a fair evaluation.

Figure 5A illustrates that LR-GCN outperforms other GNNs in terms of BLEU on the Camrest and MultiWoZ2.1 datasets, but its score is comparatively lower on the In-Car Assistant dataset. In Fig. 5B, LR-GCN exhibits a slightly higher Entity F1 score compared to other GNNs on the Camrest and MultiWoZ2.1 datasets, and significantly outperforms them on the In-Car Assistant dataset. Overall, while different GNNs offer unique advantages for specific datasets, our LR-GCN approach demonstrates the most substantial cumulative improvement in two evaluation metrics across all three datasets. We attribute this observation to LR-GCN’s utilization of low-rank matrix weight factorization to prevent overfitting and potentially better capture the global characteristics of the entire graph.

Figure 5 Performance comparison of Kg-Plug with representative GNNs.

(A) Comparison results in BLEU. (B) Comparison results in Entity F1.

Case study

Figure 6 displays the responses of PluDG along with multiple baseline models over 3 rounds of the In-Car Assistant dataset. In the first round, PluDG accurately answered the question, but with less romance compared to the ground truth. In contrast, DialoKG’s response was slightly less satisfactory, while Fg2Seq provided a more comprehensive, yet mechanical, reply. In the second round, DialoKG barely met the expectations of the ground truth. Conversely, Fg2Seq mechanically responded to the first-round responses. On the other hand, PluDG offered nearly correct answers and generated smoother, more engaging responses. In the third round, it appears that all three models responded similarly.

Figure 6 PluDG and the two baselines generated responses using the In-Car assistant dataset.

Overall, the responses generated by PluDG are more contextually appropriate and comprehensible to humans. Combining these three cases, despite the remaining gap between the sentences generated by PluDG and the Reference Entities of real responses, the first two cases demonstrate PluDG’s capability to generate semantically similar responses and provide more informative replies.

Discussion

The research results here show that the plug-in we designed can provide effective prompt entities for the decoder. After adding modules such as the Kg-Plug, the model has been greatly improved on the three data sets. Therefore, in the future, we can design some plug-and-play lightweight plug-ins to assist large language models in different domain knowledge areas to generate results. Furthermore, we believe that, apart from enhancing the reply’s accuracy, exploring ways to enhance its engagement and amusement could be a valuable area for future research, as evidenced by the results of this case study.

Conclusions

In this article, we introduce a novel task-oriented dialog system called PluDG, which utilizes a plug-and-play plug-in named Kg-Plug to assist GPT-2 in extracting knowledge base features. To enable GPT-2′s full exploration of the internal relationship of the selected knowledge base, we propose Unified Memory Integration, a method that enhances the comprehension of the sentence’s internal structure and optimizes the knowledge base encoding location, thus improving the accuracy and fluency of the responses. Our experiments on three standard datasets demonstrate that our proposed model surpasses existing state-of-the-art models, particularly on datasets with complex knowledge base information. Additionally, we perform further ablation experiments to investigate the contribution of each module to the overall model. We aspire that our research findings will make a valuable contribution to the domain of task-oriented dialogue systems.

Supplemental Information

Supplemental Information 1 Code and data

Click here for additional data file.

Additional Information and Declarations

Competing Interests

Author Contributions

Data Availability

The authors declare there are no competing interests.

Xuelian Dong conceived and designed the experiments, performed the experiments, analyzed the data, performed the computation work, prepared figures and/or tables, authored or reviewed drafts of the article, and approved the final draft.

Jiale Chen conceived and designed the experiments, performed the experiments, analyzed the data, performed the computation work, prepared figures and/or tables, authored or reviewed drafts of the article, and approved the final draft.

The following information was supplied regarding data availability:

The data and code are available in the Supplemental File.

The data used in this study are from Wen et al., 2016, EricManning, 2017, and Budzianowski et al., 2018. They are available at:

- https://paperswithcode.com/dataset/wizard-of-oz

- https://nlp.stanford.edu/blog/a-new-multi-turn-multi-domain-task-oriented-dialogue-dataset/

- https://paperswithcode.com/dataset/multiwoz

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
