# Peer review of "PluDG: enhancing task-oriented dialogue system with knowledge graph plug-in module"

_PeerJ Computer Science, doi:10.7717/peerj-cs.1707_

## Round 0.1 · original submission · Major Revisions

Please revise your manuscript to address the comments from the two reviewers, especially to add more details about the datasets and the evaluation metrics, perform a statistical test, and provide some missing information about the equations and the experiments.

Reviewer 1 ·

Basic reporting

The content that is presented is understandable, and the language that is used in the manuscript is good.
The figures are of a high quality, and detailed descriptions are provided for each one.
The article provides a sufficient amount of background information as well as an introduction to demonstrate how the work fits into the larger field of knowledge. The appropriate references are provided for any prior literature that is applicable.

Experimental design

The article clearly defines the research question, and the experiments sound good.
However, there are a few points that need to be explained in greater detail:
- What does Emb stand for in Equation (2)?
- After Equation (5), there is a typo; it should be "model" not "mode".
- What kind of training did the model receive? When the model is being trained, what hyperparameters are being used? What about the values that the authors have chosen for those hyperparameters?
- How about the computing platform that was used to train the model and/or carry out the experiments?

Validity of the findings

The findings presented in the manuscript have been verified by the results of the experiments. The conclusions are understandable and provide an answer to the original research question.

Additional comments

This manuscript requires a major revision in order to take into account the comments above.

Reviewer 2 ·

Basic reporting

- In the manuscript, the authors have utilized language that is easy to understand and relatively specific.
- The main technical content can be understood easily.
- A good visual support is offered by the figures included in the manuscript. The resolution of the figures is high enough that readers will be able to make out the essential particulars. The descriptions and labels of each figure are written in an understandable and comprehensive manner.
- The authors have provided nearly all of the results that are relevant to their hypothesis in their presentation. This indicates that the subject has been discussed in great detail. Nevertheless, the quality of the manuscript will be elevated if the authors add additional specifics, as will be discussed in the Additional comments that follow.

Experimental design

- The study aligns with the aims and scope of the journal.
- The methodology for the experiment is adequate. On the other hand, as will be pointed out below, additional information, such as that regarding the datasets, should to be included.
- Both the code and the data were provided by the authors. Those who want to extend this research or reproduce the results that have been reported would find this to be helpful.

Validity of the findings

- In my opinion, the findings of this research can be applied in the real world.
- The conclusions presented in the manuscript are articulated clearly and can be connected to the primary research question.

Additional comments

- Because the primary contributions made by this research are not discussed in sufficient detail in the manuscript, the authors should provide a summary of those contributions.
- Although the authors have included in the manuscript the sources of the datasets that were used in the experiments that they conducted, it would be beneficial if more information about those datasets was provided.
- Because BLEU and Entity F1 are not typical metrics, the authors should provide additional information regarding both of these metrics.
- The authors should conduct a statistical test in order to demonstrate that the performance of their proposed method is better than that of other methods.

---

## Round 0.2 · Minor Revisions

Please check the comments from the two reviewers and revise the manuscript accordingly, especially to address the following:
- Improve the list of references as suggested by Reviewer 1.
- Discuss the contributions of the core modules as suggested by Reviewer 2.

Reviewer 1 ·

Basic reporting

OK, no further comments.

Experimental design

The revised version has been improved. The authors have fixed the errors in the equations and addressed other questions of mine about the experiments.

Validity of the findings

OK, no further comments.

Additional comments

I am happy that the authors have addressed all of my concerns. One more additional and very minor comment is that the list of references may need to be improved as some items are on arXiv but their URLs or IDs are not included.

Reviewer 2 ·

Basic reporting

The quality of presentation and language is good, and I have no further comments.

Experimental design

The experiment design sounds good. The study aligns with the aims and scope of the journal.
The revision is improved, and the authors have already included information about the datasets, experimental settings, and evaluation metrics, as suggested in my previous report. A statistical test has been conducted as well.

Validity of the findings

The findings are valid and useful. I have no further comments.

Additional comments

Basically, the authors have already addressed all of my comments in the previous report. I still have one more comment: it is not very clear about the contributions of the core modules, i.e., Kg-Plug and UMI, to the final performance of the system. Which one is more important than the other? Do we need to include both? It would be very interesting and useful if an ablation study about this matter could be conducted. However, if this is difficult, then a discussion should be sufficient, as the performance of the proposed system is already good.

---

## Round 0.3 · accepted · Accept

The authors have addressed all of the reviewers' comments. I have reviewed the updated manuscript and believe the authors have effectively incorporated the suggestions. This version does not need to be sent to the reviewers. The manuscript is now ready for publication as a result of this evaluation.